# Would I Lie To You? Inference Time Alignment of Language Models using Direct Preference Heads

**Avelina Asada Hadji-Kyriacou**
Department of Computer Science
University of St Andrews
College Gate, St Andrews, KY16 9AJ
lhk3@st-andrews.ac.uk

**Ognjen Arandjelović**
Department of Computer Science
University of St Andrews
College Gate, St Andrews, KY16 9AJ
oa7@st-andrews.ac.uk

## Abstract

Pre-trained Language Models (LMs) exhibit strong zero-shot and in-context learning capabilities; however, their behaviors are often difficult to control. By utilizing Reinforcement Learning from Human Feedback (RLHF), it is possible to fine-tune unsupervised LMs to follow instructions and produce outputs that reflect human preferences. Despite its benefits, RLHF has been shown to potentially harm a language model's reasoning capabilities and introduce artifacts such as hallucinations where the model may fabricate facts. To address this issue we introduce *Direct Preference Heads* (DPH), a fine-tuning framework that enables LMs to learn human preference signals through an auxiliary reward head without directly affecting the output distribution of the language modeling head. We perform a theoretical analysis of our objective function and find strong ties to Conservative Direct Preference Optimization (cDPO). Finally we evaluate our models on GLUE, RACE, and the GPT4All evaluation suite and demonstrate that our method produces models which achieve higher scores than those fine-tuned with Supervised Fine-Tuning (SFT) or Direct Preference Optimization (DPO) alone.

## 1 Introduction

Reinforcement Learning from Human Feedback (RLHF) is a technique that can be used to align an agent — such as a Large Language Model (LLM) — to human preferences and lead to more truthful, more helpful, less harmful and more preferred outputs [29]. Proximal Policy Optimization (PPO) [36] and Direct Preference Optimization (DPO) [31] are two such alignment techniques which have been extensively used to improve the quality of LLM outputs, leading to instruction following agents or chat assistants which are quickly approaching human-baselines in a variety of knowledge and reasoning tasks [4, 10, 42, 18, 24, 35, 11].

However, recent research has shown that RLHF may actually hurt an LLM's reasoning abilities rather than improve it. One study [5] discovered that performing alignment during the Supervised Fine-Tuning (SFT) stage of training may lead to worse performance on reasoning benchmarks, and another [3] that SFT alone outperforms RLHF for smaller models with the benefits of RLHF only emerging for models with more than 1 billion parameters. Ouyang et al. [29] also report an increased tendency for RLHF models to make up information in closed domain tasks ("hallucination") compared to models trained with SFT alone.

To combat the the risk of RLHF compromising the abilities of an LLM in favour of producing preferable outputs we introduce Direct Preference Heads (DPH), a novel feature based approach that optimises a reward score produced by the LLM rather than optimising the logits produced by language modelling head. DPH can be used in combination with (or without) existing alignment

38th Conference on Neural Information Processing Systems (NeurIPS 2024).

techniques to allow language models to self-evaluate outputs sampled at inference time and select the highest scoring candidate.

We evaluate the performance of DPH using an efficient 551M parameter LM on a variety of commonsense reasoning and Natural Language Understanding (NLU) tasks. All code used to train our models is available on GitHub[1] and we release our model weights on Hugging Face[2].

## 2 Prior Approaches

Prior approaches to language model alignment involve directly optimizing the logits produced by the language modelling head to increase the likelihood of producing preferable responses while decreasing the likelihood of undesirable responses.

### 2.1 Reinforcement Learning from Human Feedback (RLHF)

Reinforcement Learning from Human Feedback seeks to learn a reward model from human feedback on completions generated by a language model which can be used to align an LM with human preferences. A typical RLHF pipeline consists of 3 steps: (1) supervised fine-tuning, (2) preference sampling and reward modelling, and (3) RL fine-tuning.

**Supervised Fine-Tuning** The first step of a standard RLHF pipeline is fine-tuning a pre-trained LM on high quality data for downstream tasks to obtain a model $\pi^{\text{SFT}}$.

**Reward Modelling** Next, the SFT model is prompted with input tokens $x$ to produce completions $y$. These answers are then rated by human labellers which rate the answers based on one or more criteria. A reward model $r_\phi(x, y)$ is then trained to estimate the scores assigned by human labellers using maximum likelihood estimation.

**RL Fine-Tuning** During the RL phase the learned reward function is used to provide feedback to the language model using the following optimization problem

$$\max_{\pi_\theta} \mathbb{E}_{x \sim \mathcal{D}, y \sim \pi_\theta(y|x)} \left[ r_\phi(x, y) \right] - \beta D_{\text{KL}} \left[ \pi_\theta(y|x) || \pi_{\text{ref}}(y|x) \right], \tag{1}$$

where $\beta$ controls the deviation from the base reference policy $\pi_{\text{ref}}$, which is typically initialized from $\pi^{\text{SFT}}$. Due to the non-differentiable nature of language generation this objective must be optimized using a reinforcement learning algorithm such as PPO [36].

### 2.2 Direct Preference Optimization (DPO)

Direct Preference Optimization was introduced as a reparametrization of RLHF which eliminates both the sampling stage and the reward modelling stages and reformulates alignment procedure as a loss function which can be optimized directly on a dataset of pairs of preferred and dispreferred completions to given prompts. This allows DPO to stably and efficiently converge on an optimal policy using what is effectively a classification loss over positive and negative pairs.

Given a dataset $\{(x, y_w, y_l)\}$ where $x$ is the prompt and $y_w, y_l$ are the preferred and dispreferred completions, we introduce the following loss function:

$$\mathcal{L}_{\text{DPO}}(x, y_w, y_l) = -\log \sigma \left( \beta \log \frac{\pi_\theta(y_w|x)}{\pi_{\text{ref}}(y_w|x)} - \beta \log \frac{\pi_\theta(y_l|x)}{\pi_{\text{ref}}(y_l|x)} \right), \tag{2}$$

where $\pi_\theta(y_*|x)$ and $\pi_{\text{ref}}(y_*|x)$ are the probabilities of completions $y_*$ for prompt $x$ given by the policy model and reference models respectively, and the $\beta$ parameter controls the deviation from the reference policy.

There also exists an augmentation of DPO namely Conservative DPO (cDPO) [26] which is designed to be more robust to noisy labels through the introduction of label smoothing parameter $\epsilon$. The objective function for cDPO is given by:

$$\mathcal{L}_{\text{cDPO}}(x, y_w, y_l) = (1 - \epsilon)\mathcal{L}_{\text{DPO}}(x, y_w, y_l) + \epsilon\,\mathcal{L}_{\text{DPO}}(x, y_l, y_w). \tag{3}$$

---

[1] https://github.com/Avelina9X/direct-preference-heads

[2] https://huggingface.co/collections/Avelina/direct-preference-heads-preprint-6612d8a6fa3843352943fd43

# 3 Direct Preference Heads

The hypothesis underlying the Direct Preference Optimization framework of Rafailov et al. [31] is that a "language model is secretly a reward model" thereby making the purpose of Direct Preference Heads to exploit this and extract explicit reward signals without the need of an *additional* reward model.

## 3.1 Reward Head

To obtain the rewards from a sequence $x; y$ three components are required: an aggregated hidden state $h$ which is conditioned on the intermediate representations of the language model, a pooling function $f$ which transforms the hidden state, and a learnable vector $w_{dph}$ with the same dimension as the output of $f$. We then compute the reward $r$ as follows:

$$r = f(h) \cdot w_{dph}. \tag{4}$$

To obtain the hidden state we take the output of the last transformer layer for the final token of the sequence, and we experiment with three choices of $f$: (1) the identity mapping following the convention established by OpenAI's GPT for sequence classification [30], (2) a learnable affine projection with $\tanh$ nonlinearity following BERT's pooling function [21], and (3) an inverted bottleneck FFN with SwiGLU activation mirroring the FFN blocks used within the transformer backbone followed by $\tanh$ nonlinearity [37].

## 3.2 Objective Function

We formulate two novel objective functions for our method: a separable objective which maximises positive rewards and minimises negative rewards, and a contrastive objective which maximises the margin between positive and negative rewards. The loss landscapes are illustrated by Figure 1 in the appendix.

### 3.2.1 Separable DPH

The Separable DPH loss function given by (5) is a function of the preferred and dispreferred rewards $r_w, r_l$, and the label smoothing parameter $0 \leq \epsilon \leq 0.5$ which controls the reward margin:

$$\mathcal{L}_{\text{SepDPH}}(r_w, r_l) = - \left[ (1-\epsilon) \log \sigma(r_w) + \epsilon \log \sigma(-r_w) \right] - \left[ \epsilon \log \sigma(r_l) + (1-\epsilon) \log \sigma(-r_l) \right]. \tag{5}$$

**Theorem 1.** *For all $\epsilon \in (0, 0.5]$ the objective function $\mathcal{L}_{SepDPH}$ is convex and will optimize the policy $\pi_\theta$ such that the preferred rewards $r_w$ produced by the preference head converge towards $\log \frac{1-\epsilon}{\epsilon}$ and the dispreferred rewards $r_l$ converge to $\log \frac{\epsilon}{1-\epsilon}$.*

This can be proven by observing the first and second partial derivatives of the loss function with respect to the rewards. The first partial derivative is equal to zero at the points $r_w = log \frac{1-\epsilon}{\epsilon}$ and $r_l = \log \frac{\epsilon}{1-\epsilon}$ respectively, and the second partial derivative is strictly positive for all values of $r_w, r_l$. A full proof is included in Appendix A.1.

### 3.2.2 Contrastive DPH

Like Separable DPH, the loss function for Contrastive DPH given by (6) is function of the preferred and dispreferred rewards $r_w, r_l$ and the label smoothing parameter $0 \leq \epsilon \leq 0.5$. This version of the loss function optimizes the *relative* margin between the rewards rather than optimizing the *absolute* positive and negative rewards as in Separable DPH.

$$\mathcal{L}_{\text{ConDPH}}(r_w, r_l) = -(1-\epsilon) \log \sigma(r_w - r_l) - \epsilon \log \sigma(r_l - r_w). \tag{6}$$

**Theorem 2.** *For all $\epsilon \in (0, 0.5]$ the objective function $\mathcal{L}_{ConDPH}$ is convex and will optimize the policy $\pi_\theta$ such that the difference between preferred rewards $r_w$ and dispreferred rewards $r_l$ produced by the preference head will converge to a fixed margin, given by $r_\Delta = r_w - r_l = \log \frac{1-\epsilon}{\epsilon}$.*

This can be proven by reparametrising the loss function such that $r_\Delta = r_w - r_l$ and then by considering the first and second partial derivatives with respect to this reward margin. It can be observed that the first partial derivative is equal to zero when $r_\Delta = \log \frac{1-\epsilon}{\epsilon}$, and the second partial derivative is strictly positive for all values of $r_\Delta$. A full proof is included in Appendix A.2.

### 3.2.3 Relation to cDPO

The properties of both Contrastive DPH and Separable DPH show a strong relationship with Conservative DPO: SepDPH will converge to optimal *fixed reward margins* above zero for $r_w$ and below zero for $r_l$; ConDPH will converge to optimal *fixed reward margins* between $r_w$ and $r_l$, and cDPO will converge to a *fixed delta from the reference model* [26]. Like Conservative DPO, this makes both Separable DPH and Contrastive DPH robust to preference label noise and makes training more stable than naive maximum likelihood estimation without label-smoothing.

## 3.3 Novelty over Traditional Reward Modelling

Although similar to the reward modelling phase of an RLHF pipeline, DPH has some distinct differences which set it apart. DPH does not require an SFT sampling and human labelling stage meaning it can take advantage of pre-constructed preference datasets such as those used for DPO. Typical RLHF also requires multiple models — a reward model, a reference model and a policy model — while DPH requires only a single model to produce both responses and rewards.

Unlike other RLHF pipelines such as PPO [36], the rewards produced by DPH are not used for RL fine-tuning; instead, the DPH rewards are to be used to prune candidate generations sampled from the LM at inference time to select the candidate which aligns most with human preferences. This makes DPH an excellent choice for small language models which are (1) more lightweight — and therefore can be efficiently used to generate multiple samples — and, (2) are more prone to degradation when aligned using typical RL techniques [5, 3].

# 4 Experimental Setup and Data

## 4.1 Datasets

We make use of a variety of datasets for fine-tuning and evaluation which are outlined below. The specific prompt templates used for fine-tuning and evaluation are described in Appendix C.

**Natural Language Understanding (NLU)**  For general NLU we make use of the standard **GLUE** benchmark [40]. The overall score for GLUE is computed by the macro-average of unweighted metric averages for all 9 tasks, however we also include a secondary score which does not included the 'problematic' WNLI task following the evaluation used for BERT [21]. We opted to omit WNLI during fine-tuning due to the low sample size.

**Commonsense Reasoning**  In accordance with the **GPT4All** [1] evaluation suite, we use the following datasets to evaluate commonsense reasoning abilities: **HellaSwag** [42], **OpenBookQA** [25], **WinoGrande** [35], **ARC** [10], **BoolQ** [9], and **PIQA** [7].

**Reading Comprehension**  To evaluate reading comprehension abilities we use the **RACE** dataset [22], a multiple-choice task which requires reasoning over provided passages.

**Instruction Following**  We include the **Alpaca** [38], **OpenOrca** [23], and **UltraFeedback** [12] datasets to train our models for instruction following. We make use of OpenOrca and a cleaned version of Alpaca for SFT, and binarized versions of OpenOrca and UltraFeedback for alignment.

**Auxiliary Datasets**  To provide additional training data for SFT we include the **MMLU** [18], **SQuAD V2** [33, 32], **Tiny Stories** [15], **CNN-Dailymail** [27] and **CoQA** [34] training splits. For alignment we only include MMLU and SQuAD V2.

## 4.2 Prompts and Sampling

**Prompts**  We make use of the ChatML prompt templating scheme [28] with handcrafted `system`, `user` and `assistant` prompts specific to each task. During fine-tuning we mask out the loss for all tokens of the prompt and condition the model on the content of `assistant` messages including the final `<|im_end|>` token. During evaluation we select the highest scoring answer using the average log-probabilities of the tokens in the final `assistant` message, or compute the reward scores on the final `<|im_end|>` token when evaluating with DPH.

**SFT Sampling** When sampling from the datasets for SFT we randomly shuffle each dataset and uniformly interleave samples from all tasks in the mix. To control the weighting of samples from each task we fill the context window with $n$ consecutive samples from the same task before sampling from a different task, where $n$ is chosen to be 5 in our experiments. To maximise compute utilisation and minimize unused portions of the context window we make us of Transformer-XL [13] style training with a context window size of 2048 tokens and a recurrent memory size of 2048 tokens.

**DPH Sampling** When sampling from datasets for DPH alignment we switch from the Transformer-XL style pipeline to typical SFT training, opting to only include single samples in the context window padded to a fixed maximum length. As some of the datasets we use for DPH are intended for SFT rather than alignment (namely GLUE, GPT4All, RACE, MMLU and SQuAD) we synthesise preference pairs where the 'correct' answer is used as the preferred completion and we uniformly sample an 'incorrect' answer from the available choices for the dispreferred completion. This is trivial for most datasets, however we use a special process for the SQuAD V2 dataset; for answerable questions we use "unanswerable" as the dispreferred completion, and for unanswerable questions we use SpaCy to randomly sample a noun span from the context to use as the dispreferred completion.

## 4.3 Regularization

The hidden states $h$ used to compute the reward scores are likely sub-optimal for computing rewards when initialising $\pi_\theta$ from $\pi^{\text{SFT}}$. As such, it may be desirable to fine-tune some or all parameters in the language model to learn better reward signals. This necessitates the use of regularization to prevent degradation of the models generative capabilities while learning to predict rewards.

**Prior Regularization** Typical parameter regularization strategies such as weight decay make the assumption that parameters $\theta$ follow a zero-mean Normal distribution $p(\theta) \sim \mathcal{N}(0, \frac{1}{\beta}\mathrm{I})$ leading to an auxiliary loss term $\frac{\beta}{2}||\theta||_2^2$. However, when performing transfer-learning or fine-tuning on a pre-trained model this assumption can be harmful and aid in catastrophic forgetting of the model's previously learnt abilities.

An alternative regularization scheme is Prior Regularization [8, 14, 17] which instead makes the assumption that the fine-tuned parameters are normally distributed around the original parameters $\theta_{\text{ref}}$, that is $\theta \sim \mathcal{N}(\theta_{\text{ref}}, \frac{1}{\beta}\mathrm{I})$, leading to the auxiliary loss term $\frac{\beta}{2}||\theta - \theta_{\text{ref}}||_2^2$.

We employ Prior Regularization to limit the divergence of $\pi_\theta$ from $\pi^{\text{SFT}}$ while still facilitating the learning of improved hidden state representations for the Direct Preference Head. Pseudocode for optimizer based decoupled prior regularization is included in Appendix B.1.

**cDPO Regularization** Rather than directly employing a KL divergence penalty similar to that used in (1) we find that it is possible — and even beneficial — to use Conservative DPO as a means of limiting the divergence of the policy model to a fixed delta from the reference model, and 'nudging' the model towards generating more preferable outputs which increases the chance of generating a better candidate completion at inference time with fewer sampling steps.

## 4.4 Training Pipeline

We progressively fine-tune the models in 3 stages: vocab extension, supervised fine-tuning, and DPH alignment. The details of the pre-trained model are included in Appendix D.1.

**Vocab Extension** Since our model was pre-trained without a chat structure it is necessary to train the embeddings for additional `<|im_start|>` and `<|im_end|>` tokens: we freeze all non-embedding parameters and use the same datasets as SFT. We fine-tune the embeddings for 4096 steps with a batch size of 128, a max LR of 6e-5 which warms up over 200 steps followed by cosine decay down to zero, and clip the global gradient norm to 1.

**Supervised Fine-Tuning** After vocab extension we move onto the SFT step which conditions the model for NLU tasks and instruction following using the sampling and loss masking method described in Section 4.2. We fine-tune the model for 6144 steps with a batch size of 128, a max LR of 3e-5 which warms up over 200 steps followed by cosine decay down to zero, prior-regularization applied to all non-embedding parameters with coefficient 0.5, and clip the global gradient norm to 1.

**DPH Alignment** Using the sampling method described in section 4.2 we jointly learn DPH rewards and perform cDPO alignment. The goal here is to gently push the model towards producing preferable outputs without compromising the model's reasoning abilities, and the priority is to attain the highest validation metrics from the DPH rewards. This requires balancing the two objectives, and as such we introduce weighting parameters $\alpha_1, \alpha_2$ to our final joint objective in (7) where $\mathcal{L}_{\text{DPH}}$ is either $\mathcal{L}_{\text{sepDPH}}$ or $\mathcal{L}_{\text{conDPH}}$. We find $\alpha_1, \alpha_2 = 1$ to be a good balance between DPO and DPH in our experiments.

$$\mathcal{L}_{\text{joint}}(x, y_w, y_l, r_w, r_l) = \alpha_1 \mathcal{L}_{\text{cDPO}}(x, y_w, y_l) + \alpha_2 \mathcal{L}_{\text{DPH}}(r_w, r_l) \tag{7}$$

We align the model for 23,040 steps with a batch size of 64 pairs, a max LR of 3e-6 which warms up over 200 steps followed by cosine decay down to 3e-7, prior-regularization applied to all parameters with coefficient 0.5, and clip the global gradient norm to 1. Following the optimal DPO parameters for OpenHermes-7b-2.5 [20] we use $\beta = 0.6$ and chose cDPO $\epsilon = 0.25$ and DPH $\epsilon = 0.1$ for regularisation. Additionally, we apply dropout with $p = 0.1$ to the outputs of the pooler.

## 4.5 Compute Resources

All fine-tuning was performed using an NVIDIA A100 SXM4 80GB GPU on a compute cluster, with jobs allocated 24 cores and 160GB of memory. Each checkpoint is saved in FP16 format which consumes about 1.1GB of storage; and the datasets require minimal storage space.

For vocab extension we train for 4096 steps with an average of 7.99 seconds of compute per step which translates to about 9 hours. For supervised fine-tuning we train for 6144 steps with an average of 9.26 seconds of compute per step which translates to about 16 hours. For DPH alignment we train for 23040 steps with an average of 7.21 seconds of compute per step which translates to about 46 hours. The DPH ablations with our models use about 140 hours of compute, and the Qwen ablations use about 60 hours of compute. In total, we used approximately 270 hours of A100 compute to train our models and collect the results included in our paper. We used additional compute for preliminary tests and fixing bugs for silently failing experiments although this wasn't tracked.

# 5 Results

## 5.1 Evaluation Methodology

As described in Section 4 we use NLU, commonsense reasoning and reading comprehension tasks to measure model capabilities, while the instruction following and auxiliary tasks are used to provide additional training signals. For the NLU tasks we evaluate on the test set of GLUE, providing average scores both with and without WNLI. For reading comprehension we evaluate on the RACE test set. For commonsense reasoning we follow the LM Evaluation Harness [16] implementations of these tasks, evaluating on the test sets of ARC and OpenBookQA and the validation sets of HellaSwag, WinoGrande, BoolQ and PIQA, which brings our evaluations in line with other models.

For vocab extension and SFT checkpoints we obtain model predictions from the completions with the highest scoring log-probabilities. For the DPH checkpoints we report metrics for both log-probability predictions (Ours$_{\text{DPO}}$) and predictions chosen from the DPH rewards (Ours$_{\text{DPH}}$). We use the SwiGLU-based pooler with the separable objective function for all our experiments as we found this combination to perform best overall as shown in Section 5.2.1.

### 5.1.1 Natural Language Understanding

Our results for NLU performance are included in Table 1. Note that the results for GPT-1 [30] and BERT [21] are from sub-task specific fine-tunes.

It is unsurprising that our model does not outperform BERT$_{\text{Large}}$ even though it has more parameters; this is likely due to BERT's task specific fine-tunes in comparison to our model which was jointly trained on several tasks. Despite this our instruction following DPH model achieves a 2.2% higher average GLUE score compared to task-specific GPT-1 fine-tunes and manages to attain the highest overall accuracy and macro-average on RTE and STS-B respectively.

Table 1: Comparison of GLUE performance. Dashes represent unpublished results. Note that the Spearman correlation for Ours$_{Vocab}$ is misleading and caused by predicting "0" for all test samples.

| System | Tokens | Params | MNLI m/mm | QQP F1/Acc | QNLI Acc | SST-2 Acc | CoLA M Corr | STS-B P/S Corr | MRPC F1/Acc | RTE Acc | Score w/o WNLI | WNLI Acc | Score w/ WNLI |
|---|---|---|---|---|---|---|---|---|---|---|---|---|---|
| Ours$_{Vocab}$ | 100B | 551M | 34.1/34.7 | 28.2/42.9 | 50.2 | 58.0 | 0.9 | -0.9/99.2 | 69.4/57.4 | 50.9 | 42.8 | 34.9 | 41.9 |
| Ours$_{SFT}$ | 100B | 551M | 73.6/75.0 | 59.1/82.8 | 81.4 | 90.8 | 22.7 | 80.6/92.4 | 80.6/75.2 | 71.4 | 72.0 | 38.4 | 68.2 |
| Ours$_{DPO}$ | 100B | 551M | 78.8/80.2 | 65.6/85.6 | 87.0 | 93.3 | 36.5 | 83.7/94.4 | 83.9/79.1 | 73.9 | 77.0 | 37.7 | 72.7 |
| Ours$_{DPH}$ | 100B | +19M | 80.0/80.6 | 65.8/85.3 | 87.5 | 94.0 | 43.8 | **85.3/93.0** | 85.5/80.2 | **75.3** | 78.6 | 46.6 | 75.0 |
| GPT-1 | 32B | 117M | 82.1/81.4 | 70.3/ - | 87.4 | 91.3 | 45.4 | 82.0/80.0 | 82.3/ - | 56.0 | - | - | 72.8 |
| BERT$_{Base}$ | 128B | 110M | 84.6/83.4 | 71.2/ - | 90.5 | 93.5 | 52.1 | - /85.8 | 88.9/ - | 66.4 | - | - | 78.3 |
| BERT$_{Large}$ | 128B | 340M | **86.7/85.9** | **72.1/89.3** | **92.7** | **94.9** | **60.5** | 87.6/86.5 | **89.3/85.4** | 70.1 | **82.5** | **65.1** | **80.5** |

### 5.1.2 Commonsense Reasoning

Our results for commonsense reasoning are summarized in Table 2. Note the Pythia [6] and TinyLlama [43] models were not fine-tuned for any specific task but received significantly more pre-training and have much higher parameter counts.

Table 2: Comparison of accuracy on the GPT4All test suite.

| System | Tokens | Params | HellaSwag | OpenBookQA | WinoGrande | ARC-Challenge | ARC-Easy | BoolQ | PIQA | Average |
|---|---|---|---|---|---|---|---|---|---|---|
| Ours$_{Vocab}$ | 100B | 551M | 36.93 | 28.60 | 51.14 | 26.19 | 25.67 | 61.25 | 65.39 | 42.17 |
| Ours$_{SFT}$ | 100B | 551M | 42.59 | 45.20 | 55.01 | 35.84 | 47.01 | 76.24 | 69.37 | 53.04 |
| Ours$_{DPO}$ | 100B | 551M | 44.83 | 52.40 | 57.38 | 39.76 | 53.54 | **79.08** | 72.36 | 57.05 |
| Ours$_{DPH}$ | 100B | +19M | **59.36** | **57.40** | 59.12 | **41.21** | **56.82** | 78.81 | 68.77 | **60.21** |
| Pythia-1.0B | 300B | 1.1B | 47.16 | 31.40 | 53.43 | 27.05 | 48.99 | 60.83 | 69.21 | 48.30 |
| Pythia-1.4B | 300B | 1.5B | 52.01 | 33.20 | 57.38 | 28.50 | 54.00 | 63.27 | 70.95 | 51.33 |
| TinyLlama | 3T | 1.1B | 59.20 | 36.00 | **59.12** | 30.12 | 55.25 | 57.83 | **73.29** | 52.99 |

With SFT alone we are able to attain comparable performance to TinyLlama using half as many parameters, and when applying DPH alignment we achieve a 7.2% increase over the TinyLlama average score and the highest accuracy in 5 of the 7 tasks.

### 5.1.3 Reading Comprehension

Our results for reading comprehension are included in Table 3. The results for GPT-1 were taken from a RACE specific fine-tune, and the results for LLaMA [39] were zero-shot without fine-tuning.

Table 3: Comparison of accuracy on the RACE test set.

| System | Tokens | Params | RACE-middle | RACE-high | Weighted Average |
|---|---|---|---|---|---|
| Ours$_{Vocab}$ | 100B | 551M | 26.0 | 24.6 | 25.0 |
| Ours$_{SFT}$ | 100B | 551M | 56.1 | 52.9 | 53.8 |
| Ours$_{DPO}$ | 100B | 551M | 65.9 | 59.8 | 61.6 |
| Ours$_{DPH}$ | 100B | +19M | **66.9** | **60.6** | **62.5** |
| GPT-1 | 32B | 117M | 62.9 | 57.4 | 59.0 |
| LLaMA 7B | 1T | 6.7B | 61.1 | 46.9 | 51.0 |
| LLaMA 13B | 1T | 13B | 61.6 | 47.2 | 51.4 |

Our SFT baseline achieves a higher average accuracy on RACE compared with the non fine-tuned LLaMa models but cannot match the accuracy of the RACE specific GPT-1 fine-tune; however after alignment our model attains a 3.5% higher average over GPT-1 while still maintaining excellent scores on other tasks using the same model weights.

## 5.2 Ablations

### 5.2.1 Pooling Head Function and Objective Choice

We ablate over the three pooling head and two objective function choices. We perform alignment for 7680 steps and report the validation scores in Table 4.

For both separable and contrastive objectives the SwiGLU pooler performs best on the three benchmarks, and for both GLUE and RACE the separable objective performs best overall. However during these experiments we discovered that contrastive DPH was achieving higher scores than

Table 4: Comparison of DPH validation scores for different objective and pooler combinations.

| Objective | Pooling Function | Add. Params | GLUE | GPT4All | RACE | HellaSwag | WinoGrande | PIQA |
|---|---|---|---|---|---|---|---|---|
| Separable | Identity | 1536 | 75.06 | 56.86 | 56.54 | 46.63 | 53.20 | 65.29 |
| Separable | BERT Style | 2.4M | 75.13 | 55.86 | 56.62 | 45.84 | 52.17 | 64.69 |
| Separable | SwiGLU FFN | 19M | **75.19** | 57.14 | **57.60** | 48.72 | 53.35 | 64.96 |
| Contrastive | Identity | 1536 | 74.99 | 57.66 | 54.09 | 50.93 | 53.83 | 66.87 |
| Contrastive | BERT Style | 2.4M | 73.91 | 57.07 | 55.89 | 49.98 | 54.62 | 67.30 |
| Contrastive | SwiGLU FFN | 19M | 74.04 | **58.28** | 55.95 | **51.38** | **55.80** | **67.57** |

separable DPH for specifically the sentence completion style tasks like HellaSwag, WinoGrande and PIQA. We hypothesise this is caused by situations where multiple completions to a given prompt may be plausible even though there is only one 'gold' answer, and as such the model benefits from maximising the relative reward margin with the contrastive objective rather than optimising absolute rewards with the separable objective.

### 5.2.2 Task Specific Heads

By taking the DPH checkpoint and freezing all backbone parameters it is possible to learn task specific heads and pooling functions for different downstream tasks at the cost of only 19M parameters per task. We train new heads for the three task groups and plot the confusion matrix of each head for each task average in Table 5. We further fine-tune for an additional 7680 steps on each task group using the same training setup as DPH alignment.

Table 5: Confusion matrix comparing validation scores for alternate heads.

| Benchmark | Baseline Head | GLUE Head | GPT4All Head | RACE Head |
|---|---|---|---|---|
| GLUE | 76.12 | **76.36** | 76.20 | 76.13 |
| GPT4All | 60.19 | 60.13 | **60.29** | 60.24 |
| RACE | 64.17 | 64.05 | **64.48** | 64.43 |

Unsurprisingly the GLUE and GPT4All heads achieve the highest scores for GLUE and GPT4All benchmarks respectively, however the GPT4All head manages to outperform the RACE head on the RACE benchmark. We hypothesise this may be due to the inclusion of multiple choice QA and reading comprehension tasks in GPT4All which may prove better training signals than the RACE training data alone.

### 5.2.3 Frozen Model Ablations

Our final experiments involve exploring the behaviour of DPH when applied to frozen language models without further fine-tuning of the original model weights. We experiment using the Qwen 1.5 model family [2] and train only the pooler and reward head weights, reporting results in Table 6. We use an identical training setup to DPH alignment but disable dropout due to the low number of trainable parameters.

Because the model backbone and embeddings remain frozen during alignment the 'Log' scores represent the model's pre-trained (or fine-tuned) capabilities. When observing the difference between the Log scores of the 0.5B Qwen models it is evident that the fine-tuning and alignment used to transform the pre-trained model into the "chat" model resulted in degraded performance across the three tasks. This phenomenon is less apparent for the 1.8B models, and actually results in higher GLUE scores for the "chat" variant of the model. This further confirms the hypothesis that alignment can harm the reasoning capabilities of smaller language models.

Table 6: Comparison of validation scores calculated using the log probabilities from the vanilla model checkpoints and reward scores produced by the trained Direct Preference Heads.

| System | GLUE Log | GPT4All Log | RACE Log | GLUE DPH | GPT4All DPH | RACE DPH |
|---|---|---|---|---|---|---|
| Qwen1.5-0.5B | 41.94 | 53.11 | 51.38 | 45.69 | 48.52 | 41.21 |
| Qwen1.5-0.5B-Chat | 39.82 | 49.70 | 50.32 | 48.99 | 49.72 | 46.90 |
| Qwen1.5-1.8B | 47.03 | 62.53 | 68.14 | 59.18 | 51.61 | 46.56 |
| Qwen1.5-1.8B-Chat | 53.85 | 61.69 | 67.47 | 62.38 | 54.47 | 53.33 |

For all models DPH is consistently able to attain higher scores on the GLUE tasks compared to the log probabilities produced by the language modelling head, but the opposite is observed for RACE which suggests the hidden states produced by the frozen backbone do not contain rich enough features for long range modelling tasks such as reading comprehension. We also observe the "chat" variants produce higher task scores for DPH than the non-chat variants which we hypothesise is a result of the authors' fine-tuning with the Chat-ML format which lead to the models' greater understanding of message structure and therefore improved hidden state aggregation for the final end message token.

When we combine these findings with those presented in Section 5.2.2, it becomes evident that the pooling function and reward head exhibit slower convergence when the model backbone is frozen. This observation further supports our hypothesis in Section 4.3, indicating that the hidden states generated by the models are are initially sub-optimal and that further fine-tuning is necessary to optimize these hidden states to achieve the best features for DPH.

# 6 Discussion

## 6.1 Future Work

As shown in the results section, DPH is capable of learning to assign higher rewards to preferred outputs and lower rewards to dispreferred outputs which implies the pooling function learns rich features with respect to prompt-completion pairs. We believe that it would be possible to also extract additional information from the output of the pooling function to detect finer grained signals such as helpfulness, humour, creativity, toxic content, etc. Such an approach has been explored by ArmoRM [41] which was first trained with several reward heads to reflect different types of preference scores and then a gating system to chose the best combination of heads based on the context. We believe this technique can be integrated into the DPH process to facilitate context-dependent re-ranking.

## 6.2 Limitations

The main benefit of DPH being its ability to perform alignment without directly effecting the model's output distribution is also its main limitation: unlike other alignment techniques which can help prevent the model generating harmful outputs, DPH is only capable of *detecting* harmful outputs. Although we do include DPO alignment in our experiments to reduce the likelihood of harmful outputs, DPH does not require such model alignment to function, which shifts the responsibility of rejecting harmful outputs to the end user or service provider.

From a computational perspective, DPH reformulates alignment as a re-ranking process, which necessitates sampling multiple candidate responses during inference. This approach requires more computational resources and memory compared to generating a single response. However, modern hardware used for LLM inference is often underutilized by smaller language models when generating singular responses in an autoregressive manner. By leveraging this unused compute capacity, we can generate multiple candidate responses in parallel when applying DPH to smaller models, without significantly increasing latency.

## 6.3 Conclusion

In this paper we introduced Direct Preference Heads, a novel form of language model alignment which is performed at inference time to prune candidate completions for a given prompt. Unlike other alignment techniques which coerce the model into generating human preference aligned outputs, DPH instead produces reward scores for candidate outputs without affecting the actual generation process and therefore avoids the issue of RLHF leading to degraded performance when applied to smaller language models. We formulated two loss functions for DPH and find strong connections to Conservative DPO, implying that DPH is robust to label noise and can be tuned to a specific confidence margin. Finally, we evaluated our methods on a number of NLU, commonsense reasoning and reading Comprehension tasks and found that DPH is able to consistently outperform both our SFT baseline and multiple publicly available language model checkpoints of varying size and training volume.

**Broader Impacts**

As with all language modelling systems we cannot guarantee all responses produced by our models are factually correct nor can we guarantee that they are safe and free from harmful content. Our work focuses on creating a system that helps filter out incorrect and harmful messages by scoring candidate outputs, but as with all alignment techniques our models may be susceptible to so-called 'jailbreaks' which can coerce the model into incorrectly assigning a higher score to less desirable content. To maximise safety DPH should be implemented alongside other safety guardrails such as Llama Guard [19] when used for publicly facing chat systems, and we intend for our provided model checkpoints to be used for reproduction of results and further research in the field of alignment.

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

## A  Appendix - Theory

### A.1  Full Proof of Theorem 1

We can prove **Theorem 1** by examining the partial gradients with respect to the rewards.

$$\frac{\partial}{\partial r_w}\mathcal{L}_{\text{SepDPH}}(r_w, r_l) = \epsilon - \frac{1}{e^{r_w} + 1} \tag{8a}$$

$$\frac{\partial}{\partial r_l}\mathcal{L}_{\text{SepDPH}}(r_w, r_l) = \frac{1}{e^{-r_l} + 1} - \epsilon \tag{8b}$$

From equations 8a and 8b we find that the partials gradients are both equal to zero at the points $r_w = log\frac{1-\epsilon}{\epsilon}$ and $r_l = \log\frac{\epsilon}{1-\epsilon}$ respectively. It is also interesting to note that $log\frac{1-\epsilon}{\epsilon} + \log\frac{\epsilon}{1-\epsilon} = 0$ which implies the positive and negative rewards will converge to an equal distance from 0.

$$\frac{\partial^2}{\partial r_w^2}\mathcal{L}_{\text{SepDPH}}(r_w, r_l) = \frac{e^{r_w}}{(e^{r_w} + 1)^2} \tag{9a}$$

$$\frac{\partial^2}{\partial r_l^2}\mathcal{L}_{\text{SepDPH}}(r_w, r_l) = \frac{e^{r_l}}{(e^{r_l} + 1)^2} \tag{9b}$$

If we derive the second derivatives for the rewards, as shown in equations 9a and 9b, we find that they are both strictly positive for all values of $r_w$ and $r_l$ which implies that Separable DPH is convex with respect to the rewards.

### A.2  Full Proof of Theorem 2

We can prove **Theorem 2** by examining the partial gradients with respect to the rewards.

$$\frac{\partial}{\partial r_w}\mathcal{L}_{\text{ConDPH}}(r_w, r_l) = \epsilon - \frac{e^{r_l}}{e^{r_l} + e^{r_w}} \tag{10a}$$

$$\frac{\partial}{\partial r_l}\mathcal{L}_{\text{ConDPH}}(r_w, r_l) = \frac{e^{r_l}}{e^{r_l} + e^{r_w}} - \epsilon \tag{10b}$$

From equations 10a and 10b we can see a symmetry emerge, where the partial gradients with respect to the preferred logits are equal and opposite to the partial gradients with respect to the dispreferred logits. If we reparameterise the loss function such that $r_\Delta = r_w - r_l$ we can derive the following partial derivative

$$\frac{\partial}{\partial r_\Delta}\mathcal{L}_{\text{ConDPH}}(r_w, r_l) = \epsilon - \frac{1}{e^{r_\Delta} + 1} \tag{11}$$

which is equal to zero for $\epsilon \in (0, 0.5]$ at the point $r_\Delta = \log\frac{1-\epsilon}{\epsilon}$.

If we derive the second derivative of the Contrastive DPH objective function with respect to the reward margin $r_\Delta$ we obtain the following formula

$$\frac{\partial^2}{\partial r_\Delta^2}\mathcal{L}_{\text{ConDPH}}(r_w, r_l) = \frac{e^{r_\Delta}}{(e^{r_\Delta} + 1)^2} \tag{12}$$

which is strictly positive for all values of $r_\Delta$, and – with respect to the reward logits – frames Contrastive DPH as a convex optimization problem with the additional properties of guaranteed convergence to a fixed margin for all $\epsilon \in (0, 0.5]$.

### A.3  Illustrative Loss Landscape

We provide an illustration of the loss landscapes to give a visual comparison of how our objective functions 'pull' rewards towards the optimal margin bounds.

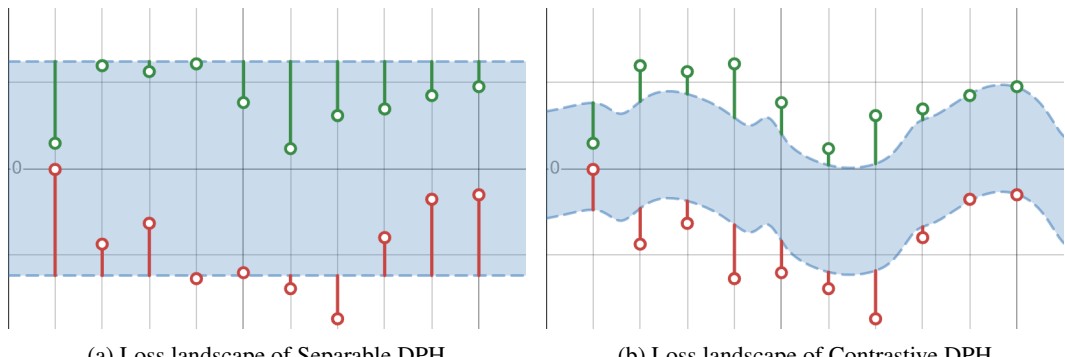

(a) Loss landscape of Separable DPH        (b) Loss landscape of Contrastive DPH

Figure 1: The loss landscapes of the DPH loss functions. The red and green points represent the rewards assigned to preferred and dispreferred answers, the vertical lines represent the direction and magnitude of reward gradients, and the blue area represents the optimal margin parameterised by $\epsilon$.

# B   Appendix - Pseudocode

## B.1   Decoupled Prior Regularization

Rather than optimizing the auxiliary loss term $\frac{\beta}{2}||\theta - \theta_{\text{ref}}||_2^2$ we can follow the procedure of decoupled weight decay and implicitly include prior regularization as a step within the optimizer update function. The pseudocode for this is included below:

---
**Algorithm 1** Decoupled Prior Regularization Update Function
---

$\lambda \leftarrow$ learning rate
$\beta \leftarrow$ regularization coefficient
$\theta, \theta_{\text{ref}} \leftarrow$ current, initial parameters

---

$\theta \leftarrow \theta - \beta\lambda(\theta - \theta_{\text{ref}})$                $\triangleright$ Prior regularization step
$\theta \leftarrow$ optimizer update step             $\triangleright$ Normal optimizer update

---

# C   Appendix - Data

## C.1   Dataset Mixes

## C.2   Data Licences

| Dataset | License |
|---------|---------|
| GLUE - CoLA | No License |
| GLUE - MNLI | Multiple (OANC, CC BY-SA 3.0) |
| GLUE - MRPC | MSR-SSLA |
| GLUE - QNLI | CC BY-SA 4.0 |
| GLUE - QQP | Other |
| GLUE - RTE | No License |
| GLUE - SST-2 | No License |
| GLUE - STS-B | Multiple (CC BY-SA 3.0, CC BY-SA 4.0) |
| GLUE - WNLI | CC BY 4.0 |
| HellaSwag | MIT License |
| OpenBookQA | Apache-2.0 |
| WinoGrande | CC-BY |
| ARC | CC BY-SA 4.0 |
| BoolQ | CC BY-SA 3.0 |
| PIQA | AFL-3.0 |
| RACE | Other |
| SQuAD V2 | CC BY-SA 4.0 |
| MMLU | MIT License |
| Tiny Stories | CDLA-Sharing-1.0 |
| CNN-Dailymail | Apache-2.0 |
| CoQA | Multiple (CC BY-SA 4.0, MSR-LA, Other, Apache) |
| Alpaca Cleaned | CC-BY-4.0 |
| OpenOrca | MIT License |
| OpenOrca Binarized | Apache-2.0 |
| UltraFeedback Binarized | MIT License |

Note that three of the GLUE tasks have no license specified on their homepages nor within their publications: CoLA claims their dataset falls under "fair use," no concrete license can be found for RTE nor its pre-cursors, and SST-2 does not specify a license.

## C.3   Prompt Templates

For brevity, we only include the prompt templates of the tasks we use for evaluation. All other prompt templates are listed within the code repository.

### C.3.1   GLUE

**GLUE - CoLA**

| System | User | Assistant |
|--------|------|-----------|
| Below is an instruction that describes a task. Write a response that appropriately completes the request using the provided answer options. | Given the following sentence, answer the question with "yes" or "no". 

 Sentence: {{sentence}} 

 Question: Does this sentence make sense? 

 Answer: | {{no \| yes}} |

**GLUE - MNLI**

| System | User | Assistant |
|---|---|---|
| Below is an instruction that describes a task. Write a response that appropriately completes the request using the provided answer options. | Given a premise statement and a hypothesis statment, respond with "True" if the premise entails the hypothesis, respond with "False" if the premise contradicts the hypothesis, or respond with "Neither" if the statements are neurtral.

Premise: {{premise}}

Hypothesis: {{hypothesis}}

Question: True, False or Neither?

Answer: | {{True \| Neither \| False}} |

**GLUE - MRPC**

| System | User | Assistant |
|---|---|---|
| Below is an instruction that describes a task. Write a response that appropriately completes the request using the provided answer options. | Given the following sentences, answer the question with "yes" or "no".

Sentence 1: {{sentence1}}

Sentence 2: {{sentence2}}

Question: Do both sentences mean the same thing?

Answer: | {{no \| yes}} |

**GLUE - QNLI**

| System | User | Assistant |
|---|---|---|
| Below is an instruction that describes a task. Write a response that appropriately completes the request using the provided answer options. | Given the following sentences, answer the question with "yes" or "no".

Sentence 1: {{question}}

Sentence 2: {{sentence}}

Question: Does Sentence 2 correctly answer Sentence 1?

Answer: | {{yes \| no}} |

**GLUE - QQP**

| System | User | Assistant |
|---|---|---|
| Below is an instruction that describes a task. Write a response that appropriately completes the request using the provided answer options. | Given the following sentences, answer the question with "yes" or "no".

Sentence 1: {{question1}}

Sentence 2: {{question2}}

Question: Do both sentences ask the same question?

Answer: | {{no \| yes}} |

**GLUE - RTE**

| System | User | Assistant |
|---|---|---|
| Below is an instruction that describes a task. Write a response that appropriately completes the request using the provided answer options. | Given the following sentences, answer the question with "yes" or "no".

Sentence 1: {{sentence1}}

Sentence 2: {{sentence2}}

Question: Do both sentences mean the same thing?

Answer: | {{yes \| no}} |

**GLUE - SST-2**

| System | User | Assistant |
| --- | --- | --- |
| Below is an instruction that describes a task. Write a response that appropriately completes the request using the provided answer options. | Given the following sentence, answer the question with "positive" or "negative".

Sentence: {{sentence}}

Question: Is this sentence positive or negative?

Answer: | {{negative \| positive}} |

**GLUE - STS-B**

| System | User | Assistant |
| --- | --- | --- |
| Below is an instruction that describes a task. Write a response that appropriately completes the request using the provided answer options. | Given the following sentences, answer the question with a number between 0 and 5.

Sentence 1: {{sentence1}}

Sentence 2: {{sentence2}}

Question: On a scale of 0 to 5 how similar are Sentence 1 and Sentence 2?

Answer: | {{0 \| 1 \| 2 \| 3 \| 4 \| 5}} |

**GLUE - WNLI**

| System | User | Assistant |
| --- | --- | --- |
| Below is an instruction that describes a task. Write a response that appropriately completes the request using the provided answer options. | Given the following sentences, answer the question with "yes" or "no".

Sentence 1: {{sentence1}}

Sentence 2: {{sentence2}}

Question: Based on the information in Sentence 1, can we concluded that Sentence 2 is true?

Answer: | {{no \| yes}} |

### C.3.2 Commonsense Reasoning

**HellaSwag**

| System | User | Assistant |
| --- | --- | --- |
| Below is an instruction that describes a task. Write a response that appropriately completes the request. | Continue the following sentence:
"{{context}}" | {{ending}} |

**OpenBookQA**

| System | User | Assistant |
| --- | --- | --- |
| Below is a question, paired with multiple choices. Respond with the choice that correctly answers the question. | Question: {{question_stem}}

Choices:
{{label[0]}}. {{choice[0]}}
{{label[1]}}. {{choice[1]}}
{{label[2]}}. {{choice[2]}}
{{label[3]}}. {{choice[3]}}

Answer: | {{label}}. {{choice}} |

**WinoGrande**

| System | User | Assistant |
| --- | --- | --- |
| Below is an instruction that describes a task. Write a response that appropriately completes the request. | Continue the following sentence:
"{{sentence.prefix}}" | {{option}} {{sentence.suffix}} |

**ARC**

| System | User | Assistant |
|---|---|---|
| Below is a question, paired with multiple choices. Respond with the choice that correctly answers the question. | Question: {{question}}

Choices:
{{label[0]}}. {{choice[0]}}
. . .
{{label[n]}}. {{choice[n]}}

Answer: | {{label}}. {{choice}} |

**BoolQ**

| System | User | Assistant |
|---|---|---|
| Below is an instruction that describes a task. Write a response that appropriately completes the request using the provided answer options. | Given the following sentences, answer the question with "yes" or "no".

Background: {{passage}}

Question: {{question}}

Answer: | {{no \| yes}} |

**PIQA**

| System | User | Assistant |
|---|---|---|
| Below is an instruction that describes a task. Write a response that appropriately completes the request. | Write a solution to the following sentence: "{{goal}}" | {{solution}} |

### C.3.3 Reading Comprehension

**RACE**

| System | User | Assistant |
|---|---|---|
| Below is a question, paired with a background context and multiple choices. Respond with the choice that correctly answers the question. | Background: {{article}}

Question: {{question}}

Choices:
A. {{option[0]}}
B. {{option[1]}}
C. {{option[2]}}
D. {{option[3]}}

Answer: | {{A \| B \| C \| D}}. {{option}} |

# D   Appendix - Model Details

## D.1   Pre-Trained Model

Our pre-trained model was developed in house for efficiency and takes advantage of techniques such as RoPE, SwiGLU activations and Flash Attention. The model totals 551 Million parameters (including embeddings).

We initialise the embeddings from OPT-125m and use embedding tying for the language modelling head. Since our model dimension is 1536 while the embedding dimension is 768 the model contains an up-projection as the first layer of the backbone and a down-projection for the final layer. There are a total of 18 transformer blocks in the model backbone which use pre-layer norm in the attention and FFN residuals. The attention blocks have 24 attention heads and we use RoPE with a base frequency of 500,000 for positional embedding, and the FFN block uses SwiGLU activation with an intermediate dimension of 4096. The context window of the model is 2048 tokens and the Transformer XL recurrent memory contains 2048 tokens which allows the model to use a sliding window size of up to 4096 tokens at inference without any degradation.

The model was trained for approximately 100 billion tokens on the first 24 shards of The Pile. Each batch is constructed of 480 sequences of 2048 tokens each which are continuously sampled from the datasets shards using queues for the Transformer XL style pre-training method.

We use the LaProp optimizer [44] with $\beta_1 = 0.9, \beta_2 = 0.95$, a max learning rate of 6e-4 which warms up over 2000 steps and cosine decays down to 6e-5, LR-coupled weight decay of 0.1 and global gradient clipping with a max norm of 1.

Each epoch of 256 steps takes 1 hour and 59 minutes on 4x RTX A4500 GPUs. For the full 398 epochs (or 101888 steps) this comes out to around 790 hours or just under 33 days of training time (ignoring time for validation in-between epochs and at the end of training).

