# OpenReview forum: "Would I Lie To You? Inference Time Alignment of Language Models using Direct Preference Heads"
_NeurIPS.cc/2024/Conference — NeurIPS 2024 poster_

### Official Review · Reviewer_3kiM · 2024-06-29

**Soundness:** 2
**Presentation:** 2
**Contribution:** 2
**Rating:** 3
**Confidence:** 4

**Summary:**

The paper introduces DPH, a new method for pre-trained language models that addresses the limitations of RLHF. Unlike traditional RLHF, which can compromise a model's reasoning abilities and cause hallucinations, DPH employs an auxiliary reward head to learn human preference signals without altering the LM's output distribution. The authors conduct a theoretical analysis linking their objective function to cDPO and demonstrate that DPH can be used in conjunction with existing alignment techniques to improve performance. The experiment results show that models fine-tuned with DPH achieve higher scores compared to those fine-tuned with SFT or DPO alone.

**Strengths:**

The authors highlight the side effects of RLHF, such as damage to the model's ability to reasoning and hallucinations, and try to solve them. I believe the research problem is significant and worth investigating. The authors are going to release their code and model weights, which would benefit our community.

**Weaknesses:**

1.	The claim that the auxiliary reward head avoids affecting the output distribution of the language modeling head is undermined by the practical implementation. Specifically, this claim does not hold when the backbone language model is updated to learn the preference distribution (line 180) and when the model is updated using a joint loss function (Eq. 7). These aspects of the implementation weaken the validity of their claim.
2.	Experiment setup: The baseline setup in the experiments is not reasonable. The authors compare their method with other language models that have distinct training settings and data, rather than comparing with other alignment methods. This makes it difficult to infer the superiority of the proposed method over existing alignment techniques, as the comparisons are not directly relevant.
3.	Clarity and organization: The writing is not easy to follow and requires improvement. Specifically, sections 4.3 and 4.4, which are parts of the methodology, are incorrectly placed in the experiment section. Additionally, the evaluation protocol (section 5.1) should be introduced in section 4 but is instead placed in the results section. This misplacement affects the clarity and logical flow of the paper, making it harder for readers to understand the methodology and its evaluation comprehensively.

**Questions:**

1. Why does DPH not require an SFT sampling and human labelling stage? In section 4.4, you mentioned SFT in your training pipeline.
2. What is the relationship between the proposed approach and the sentence "Would I Lie To You?" in your title?

**Limitations:**

The authors include a discussion regarding the limitations of the proposed method.

---

> ### Author Rebuttal · Authors · 2024-08-06
>
> **Addressing Weakness 1:**
> Although we do jointly train both the reward head and the preference distribution for completions this is realised through cDPO with a large beta penalty (which limits divergence of the output distribution) and a large epsilon (which limits the preferred vs dispreferred margin). So although the model does tend towards the LM head producing a more preferable output distribution this is heavily limited and was included purely to increase the chance of producing more preferable candidates. It is completely possible to swap cDPO with KL divergence to mitigate any change in the output distribution while learning the reward head weights, but - as stated in the paper - we found it beneficial to "slightly" align the model with regularised cDPO. We also do experiments using Qwen models in section 5.2.3 where the model backbone is completely frozen (meaning the output distribution cannot change) and find DPH achieves the highest scores on GLUE across the board, while lagging behind on other tasks which we believe is due to hidden state of the last token in the sequence not containing enough information for the commonsense and reading comprehension tasks - a phenomenon which would be mitigated by additionally training the backbone with either cDPO or KL. Additional ablations using KL divergence rather than cDPO would be an excellent candidate for inclusion in a paper revision either in the main body, or as additional experiments in the appendix.
>
> **Addressing Weakness 2:**
> The baselines we chose are from models of comparable size or capability with tradeoffs such as task-specific fine-tunes for smaller models or using base pre-trained checkpoints for larger models. To obtain baselines for similarly sized models using other alignment methods would have required us to train these models ourselves which requires significant time and compute. Had we compared our models to other publically available aligned checkpoints our models would have performed worse across the board as these publically available models are typically orders of magnitude larger, and even their SFT-only counterparts would outperform even our best in-house models. We aimed to show the efficacy of our method by showing how DPH improves over our LM head baselines and the baselines of other popular models.
>
> **Addressing Weakness 3:**
> We appreciate the reviewer's suggestion on organisation of sections which we will address in further revisions of the paper.
>
> **Addressing Question 1:**
> DPH does not require SFT sampling and human labelling as we do not follow the standard RLHF training pipeline and instead opt for a pipeline similar to that of DPO or ORPO which only requires binarized preference pairs for alignment, where such high quality datasets are readily available. We first must perform SFT to get a baseline policy (for both cDPO and prior regularisation) which is also necessary for other alignment methods like DPO and PPO. SFT sampling and human labelling could be incorporated into training however this is not a necessary step and would add complexity.
>
> **Addressing Question 2:**
> The sentence "Would I Lie To You?" refers to the fact that language models can often produce incorrect output which may be unhelpful or factually incorrect. We used this tag line because the model may sometimes generate "lies," but DPH is intended to rank such candidate outputs lower than more correct candidates regardless of log-probability.

---

> > ### Comment · Reviewer_3kiM · 2024-08-09
> >
> > After reading the rebuttal and other reviews, here are my additional comments (hope other reviewers are also aware of following):
> >
> > TL;DR
> > - The experimental setup is considerably flawed:
> >     - Evaluating an **alignment/preference tuning approach** using baselines with only pretraining (Table 2, Pythia & TinyLlama; Table 3 Llama) (some baselines are SFT)
> >     - Not including other preference tuning approaches as baselines
> > - Motivation is disconnected the proposed approach
> >     - Claiming RLHF compromising an LLM (authors' motivation) but updating the parameters of the backbone model by DPH
> >     - Not showing DPH does not compromise an LLM
> >
> > ---
> > ***Experimental setup is considerably flawed***
> >
> > The experimental design is deeply misguided. The proposed approach is a preference tuning (PT) algorithm that is conducted after SFT and trained by preference data. **A legitimate experimental setup would directly compare this with other preference tuning algorithms (DPO, KTO, IPO, etc.)**. However, the comparisons presented are like:
> > - Proposed approach: Base_Model_A + Pre-training_A + SFT_A + Preposed_PT
> > - Others: Base_Model_B + Pre-training_B (+ SFT_B or no SFT_B + unknown PT or no PT)
> >
> > It is **really confusing when you expect readers to evaluate a preference tuning appraoch by comparing a PT-ed model with a pre-trained model** (Table 2, Pythia & TinyLlama; Table 3 Llama).
> >
> > > To obtain baselines for similarly sized models using other alignment methods would have required us to train these models ourselves which requires significant time and compute.
> >
> > This excuse is unconvincing. I believe you only need to take SFT checkpoint of your model (551M) and conduct other PT baselines like DPO. The computational cost would not exceed your method (cDPO + DPH). Furthermore, the same comparison based on TinyLlama backbone is expected to make it "real" extensive evaluation. These are tiny models and can be done on a single A100 GPU.
> >
> > ---
> > ***Motivation is disconnected the proposed approach***
> >
> > The authors argue that RLHF compromises an LLM, and thus propose a method to update only the added reward heads, purportedly to prevent any impact on the model's output distribution (line 10). However, their practical implementations (section 4.3) are:
> > - Updating the backbone model, which is shared by language modelling head, by DPH + prior regularization
> > - Updating the backbone model by DPH and cDPO loss
> >
> > These steps clearly alter the model's output distribution, directly contradicting the claimed motivation. Thus, the superiority of the proposed approach should seek strong support from the empirical results, which is unfortunately missing.

---

### Official Review · Reviewer_LdN5 · 2024-07-03

**Soundness:** 3
**Presentation:** 2
**Contribution:** 3
**Rating:** 6
**Confidence:** 2

**Summary:**

This paper introduces Direct Preference Heads (DPH), a novel method for aligning language models (LLMs) with human preferences at inference time. DPH works by adding an auxiliary reward head to the LLM that learns to predict human preference scores for generated outputs. This allows the model to self-evaluate multiple candidate outputs and select the highest-scoring one, effectively pruning undesirable responses.

The authors argue that DPH offers several advantages over traditional Reinforcement Learning from Human Feedback (RLHF) methods like PPO and DPO:

* Inference-time alignment: DPH aligns the model at inference time, avoiding the potential degradation of reasoning abilities often observed with RLHF during training.
* Lightweight: DPH requires only a single model to produce both responses and rewards, unlike RLHF which typically involves multiple models.

The paper presents two objective functions for DPH, separable and contrastive, and demonstrates their connection to Conservative DPO. Experiments on GLUE, GPT4All, and RACE datasets show that DPH consistently outperforms both supervised fine-tuning (SFT) and DPO alone.

**Strengths:**

* Novel approach: DPH offers a new perspective on LLM alignment by focusing on inference-time pruning rather than modifying the generation process itself.
* Theoretical grounding: The paper provides a theoretical analysis of the objective functions and their relationship to cDPO, demonstrating robustness to label noise.
* Strong empirical results: DPH consistently outperforms baselines on various tasks, showcasing its effectiveness.

**Weaknesses:**

* Lack of statistical significance: The paper does not report error bars or statistical significance measures, making it difficult to assess the robustness of the results.

* Missing the comparison to standard RLHF methods: the paper does not include comparison to the PPO baseline, which is the most commonly used technique.

* The authors do not discuss other inference-time alignment techniques that exist in the literature.

**Questions:**

* The paper uses different sampling strategies for SFT and DPH. It would be helpful to discuss the rationale behind these choices and explore the impact of different sampling strategies on DPH performance.

* Sampling multiple candidate outputs at inference time can be computationally expensive, especially for larger LLMs. The paper doesn't explicitly address this cost, which could be a practical concern for real-world applications. Can you elaborate more on this in the paper?

**Limitations:**

The paper focuses on a 551M parameter model. It would be valuable to see how DPH scales to larger models, as RLHF is known to be more effective for larger models. Comparing DPH to RLHF-aligned larger models would provide a more comprehensive evaluation.

Also RLHF baselines on top of DPO need to be added.

---

> ### Author Rebuttal · Authors · 2024-08-06
>
> **Addressing Weakness 1:**
> The paper does not include statistical significance measures as this would require performing each stage of the training pipeline multiple times which is computationally costly and we opted to use our compute budget to perform ablations with different hyperparameters and using different models.
>
> **Addressing Weakness 2:**
> Comparing to a PPO baseline would have vastly increased the scope of our work as it would also involve steps such as SFT sampling, collecting human labels and training a reward model, or using pre-labelled datasets to train a reward model, and then performing one or more rounds of PPO with rejection sampling. This would add a plethora of choices in the alignment pipeline which would need to be ablated over and further add to the computational cost of the research. Our goal with this paper was to lay the foundations of DPH as a novel method which could be used as a baseline for further research.
>
> **Addressing Weakness 3:**
> There are a variety of inference-time alignment techniques, varying from prompt based approaches to chain of thought based self evaluation to activation steering. All these methods, however, are typically performed on significantly larger language models which on account of their greater size would likely outperform our models even without these ad-hoc alignment methods. An alternative would be reproducing one or more of these methods with smaller models, but that would require significant time and compute.
>
> **Addressing Question 1:**
> To maximise compute throughput we use the Transformer-XL style sampling strategy, which also allows for performing SFT on samples which are larger than the LM context window (such as multi-turn conversations) due to the recurrent memory mechanism. Additionally, sampling a task multiple times was introduced as a way to balance the rate each task was sampled from but ultimately all tasks remained balanced.
>
> However, for DPH alignment we switched to typical on task sample per sequence in the batch as the batch needed to be constructed from positive and negative sample pairs. This isn't really possible using the Transformer-XL as we need to exactly match the positive and negative pairs one-to-one which becomes difficult when packing multiple samples of varying lengths into a single sequence.
>
> **Addressing Question 2:**
> Sampling multiple candidate outputs does indeed require more compute, however we frame DPH as a method which is useful for smaller LLMs which often don't saturate compute capabilities of higher end accelerators meaning larger numbers of candidates can be computed in parallel using similar compute cost as a single candidate generated by a larger LLM. There are other situations where DPH may be applicable such as inference on the edge where inference latency may be of less concern and where model weights of larger LLMs may not even fit on device.
>
> **Addressing Limitations:**
> Although our paper focuses on our in house 550M parameter model we do perform ablations on 4 Qwen 1.5 family models in section 5.2.3.
>
> See weakness 2 for comments on RLHF comparisons. Such comparisons may be carried out in future works.

---

### Official Review · Reviewer_zurN · 2024-07-11

**Soundness:** 3
**Presentation:** 3
**Contribution:** 3
**Rating:** 6
**Confidence:** 3

**Summary:**

Paper proposes Direct Preference Heads, which learns a reward prediction head using a pretrained model, without affecting the model's output distribution.

**Strengths:**

Comprehensive evaluation
- paper presented experimental results across a wide range of tasks (NLU, commonsense reasoning, reading comprehension, etc. ), and compared the proposed method against a range of different baselines (Pretrained model, SFT, DPO, etc. )

Good presentation
- results are cleanly presented, training objectives are clear

Significance
- results showed that the proposed method outperforms comparable baselines in most presented tasks

**Weaknesses:**

Some missing/confusing parts about the proposed method:
- How are the learned reward predictions used? Some parts of the paper seems to suggest that multiple responses are sampled from the model, and the reward head is used to rank the samples? I think this is an important component of how the proposed method would be used, so it would be great to further elaborate on this procedure, e.g. how many samples?
- Some parts of the paper seems to suggest that the proposed approach only learns a reward prediction head on top of an existing model without modifying the original model (e.g. in abstract, "... without directly affecting the output distribution of the language modeling head ..."), but other parts of the paper stated that it is better to finetune the entire model when learning to predict rewards (e.g. Section 4.3). However, the stated reason for why DHP might be better than other RLHF techniques like DPO is that further alignment (past SFT) can hurt model performance. If DHP also requires further finetuning, what is the intuition for why DHP performs better than DPO?

Related works
- Paper introduces PPO, DPO and cDPO. However, there has been a large number of recent works in RLHF beyond these papers. It would be great to present a more complete related works section with a more comprehensive survey of existing work.

**Questions:**

See weaknesses

**Limitations:**

Yes

---

> ### Author Rebuttal · Authors · 2024-08-06
>
> **Addressing Weakness 1:**
> The learned reward predictions are to be used to rerank candidate competitions at inference time, similar to rejection sampling. The number of samples to rank, however, is completely situational and can be determined by factors such as compute capability, memory availability and the maximum latency deemed acceptable before serving the highest rank completion to the end user. In general, the more candidates generated the more likely it is for a higher quality response to be produced. It should be pointed out that the time needed to rank the candidates is negligible as the rewards are computed alongside generation, meaning the compute, time and memory requirements are determined by generation procedure alone.
>
> This does, however, bring up an excellent direction for further research to determine if and how an early stopping heuristic may be employed to optimise the number of candidates produced.
>
> **Addressing Weakness 2:**
> DPH can indeed be employed without fine-tuning any of the model's backbone weights, and we do include experiments performing this on 4 Qwen models in section 5.2.3. However for DPH to perform best it does require further fine-tuning model weights, and we facilitate this through the usage of both prior regularisation and regularised cDPO with a large beta penalty and epsilon value to discourage the policy model from diverging from the SFT model while learning to produce rewards.
>
> It would be possible to replace cDPO with KL divergence, for example, to achieve a similar effect while completely decoupling the learning of the reward head from any form of RLHF or alignment method. We opted to use cDPO because we found it allowed the model to improve further (within the divergence and confidence bounds set by beta and epsilon) without degradation which further increases the likelihood of generating improved candidates for reranking at inference time.

---

> > ### Comment · Reviewer_zurN · 2024-08-08
> > **Response**
> >
> > I thank the authors for the response! It would be great to incorporate the clarifications in the rebuttal into the paper.

---

### Official Review · Reviewer_ebEQ · 2024-07-28

**Soundness:** 3
**Presentation:** 3
**Contribution:** 3
**Rating:** 6
**Confidence:** 4

**Summary:**

The authors propose an inference time method to align language models with human preferences without harming the model’s reasoning abilities. The method creates an auxiliary reward head that operates during inference to score potential outputs without changing the output distribution directly. The author validates approach by comprehensive evaluations on NLU, commonsense reasoning, and reading comprehension datasets.

**Strengths:**

1. Theoretical Insight: The paper provides a theoretical analysis, connecting DPH to cDPO, and providing proofs to support the convexity and effectiveness of the proposed loss functions.

2. Novel Approach: The paper introduces a novel approach allowing for preference-aligned model fine-tuning without directly affecting the output distribution, potentially reducing negative side effects like hallucination and preserving the model’s original reasoning capabilities.

**Weaknesses:**

1. Inference Time Overhead: The Direct Preference Heads method involves using language models to generate multiple candidate responses, which must then be evaluated for selection. This process is less efficient compared to traditional fine-tuning methods due to the additional computational steps required.

2. Dependence on Initial Sample Quality: The effectiveness of the generation-then-reranking approach highly relies on the quality of the initial responses generated. If the sampled responses are of low quality, DPH is unable to enhance the output, as it can only rerank the given candidates.

3. Limited Evaluation Scope: The evaluation of the DPH method is restricted to a single model of 550M parameters. It is not clear how well the method would perform when scaled to larger models, as the results might not be consistent across different model sizes.

**Questions:**

1. The paper would be more convincing if the author applied the proposed method to a larger-sized model and demonstrated a scaling trend. The use of only one model makes the effectiveness of the approach a bit underdetermined

2. How does the proposed DPH method compare with other inference time algorithm such as rejection sampling?

3. How do you choose baselines when reporting performance? For example, Table 1 reports the model performance on BERT, an early-stage masked language model, instead of other competitive baselines.

**Limitations:**

The authors addressed all the limitations in the conclusion section.

---

> ### Author Rebuttal · Authors · 2024-08-06
>
> **Addressing Weakness 1:**
> DPH does indeed have a higher inference-time overhead than other alignment methods which aim to produce optimal completions in a zero-shot manner. However we frame DPH as being a suitable method for smaller language models which, as cited in the paper, may be harmed by other RLHF methods. The inference overhead of such smaller language models often do not not saturate the available compute or memory meaning multiple candidates can often be produced in parallel with minimal increase in latency.
>
> **Addressing Weakness 2:**
> Although the quality of produced responses depends on initial sample quality, we make use of high quality training datasets which have been used to train SOTA open-weight open-dataset language models. It is, of course, possible that all candidates generated by the language model are low quality, however through tuning generation hyper-parameters and making use of advanced sampling techniques such as typical sampling or contrastive search it is possible to coerce the model into producing a candidates with high variation from which it is likely some candidates will indeed be high quality. It is possible for DPH to be combined with other alignment methods if desired to further increase the quality of the most probable completions, but this paper primarily aims at providing the foundations for DPH as an inference time reranking method.
>
> **Addressing Weakness 3:**
> See below.
>
> **Addressing Question 1:**
> We do evaluate the proposed method with larger sized models by training a reward head and pooling function on 4 frozen Qwen 1.5 models (0.5B, 0.5B-Chat, 1.8B, and 1.8B-Chat) and report the results in section 5.2.3 and Table 6.
>
> **Addressing Question 2:**
> Rejection sampling is actually very similar to reranking with DPH, with the only difference being rejection sampling typically uses a separate reward model to evaluate the candidate completions whereas our method combines the generation and reward modelling into a single LM. Additionally, rejection sampling is often used during training to generate new candidates to improve the policy through methods such as PPO, while DPH is only intended for inference time reranking.
>
> If we were to use rejection sampling during training we would run into "reward hacking" issues where the model becomes overconfident in its reward assignments on self-generated completions which would be analogous to mode collapse in GANs; this does not mean that rejection sampling cannot be used with DPH models as the reward hacking issue will be much less prevalent if a separate reward model is used for ranking. This is, however, antithetical to the lightweight nature of our method and the intention to minimise the number of concurrent models needed to perform fine-tuning and alignment, and as such we did not explore rejection sampling and other similar RLHF methods as part of our training pipeline.
>
> **Addressing Question 3:**
> Our choice of baselines were partially dictated by the tasks themselves, and partially by the availability of results. For example with GLUE we picked BERT as it is still common to finetune a BERT model for NLU tasks which makes it an excellent choice for comparison. Additionally, although BERT has fewer parameters, the results were produced by task-specific fine-tunes which we believe makes it a valid and competitive baseline to compare against our 551M multi-task fine-tunes. We also included GPT-1 to compare our model with task-specific fine-tunes of a causal language model, which we chose due to the availability of results and its inclusion in the original BERT paper. GLUE also imposes rate-limiting on the evaluation server which adds time constraints to re-evaluating newer models on the test set.
>
> For the commonsense reasoning tasks we followed the evaluation used by TinyLlama and included TinyLlama and two Pythia models, all of which had significantly more parameters and pre-training albeit without fine-tuning.
>
> And for the reading comprehension tasks we included the results from GPT-1 to compare our model with a smaller but task-specific fine-tuned model, and the results from LLaMa 7B and 13B to compare our model with larger but non fine-tuned models.
>
> These choices may seem odd, but we wanted to include verified results from popular models which were performed externally and posted publicly to reduce evaluation bias due to factors such as prompt selection, or quirks in the evaluation pipeline which may favour certain models or token vocabularies when computing log probabilities.

---

> ### Comment · Reviewer_ebEQ · 2024-08-08
>
> Thanks a lot for the additional details you provided. However, I’m still not fully convinced about the choice of baselines included in the paper and the experimental setup. Specifically, I’m curious why larger-scale state-of-the-art models, such as LLaMA-2 7B, weren’t used for the main experimental results and compared with all baselines at that scale. The choice of smaller models makes it challenging to determine if the improvements would hold at a larger scale. I’m already leaning towards accepting the paper and will keep my score unchanged, but I hope future versions will include these considerations.

---

> > ### Author Response · Authors · 2024-08-08
> >
> > Thank you for your response. It would indeed be valuable to include results from models such as Llama 2, however as stated in my previous response we elected to use verified results which were not collected by ourselves to reduce any potential bias in our testing pipeline (such as from prompt selection, or aggregation method for log-probabilities).
> >
> > However, for some benchmarks (the GPT4All suite and RACE) it may be possible to use a standardised evaluation suite such as Eleuther AI's LM Eval Harness to obtain reasonably unbiased results, and we would have the compute necessary to do so for a future revision. Evaluating GLUE, however, would be a bit more tricky, especially since STSB is a regression tasks and we had to compromise our own model's evaluation by forcing it to chose from integer predictions.

---

### Decision · Program_Chairs · 2024-09-25

**Decision:**

Accept (poster)

**Comment:**

The approach presented is interesting -- rather than directly fine-tuning a model, it proposes to train a reward model to be used at inference time only to rerank base model outputs. In practice, this reward model is the same as the base model, which has been fine-tuned to have a reward head, but not directly on generated output sequences.

The approach requires more computation at inference time (in contrast to simple fine-tuning) and its usefulness is also constrained by inference-time compute (required to sample candidates to rerank).

The experiments focused only on several small models, but the authors argue that the approach itself is focused on improving small language models, where inference-time compute is less burdensome. I suggest that the authors refine their arguments about the approach being focused on small models, and why this justify experiments and baselines from small / older models only.

There are also not experiments that provide direct evidence that the model hasn't degraded through the fine-tuning process in training it to predict rewards for inference time.

There is no direct comparison with other standard RLHF approaches such as PPO, although there are comparisons to DPO, and authors argue that PPO would be computationally infeasible for their experiments.